# Establishing a Multidisciplinary Framework for an Emergency Food Supply System Using a Modified Delphi Approach

**DOI:** 10.3390/foods11071054

**Published:** 2022-04-06

**Authors:** Shuyu Liu, Yue Li, Shaobo Fu, Xin Liu, Tao Liu, Haojun Fan, Chunxia Cao

**Affiliations:** 1Institute of Disaster and Emergency Medicine, Tianjin University, Tianjin 300072, China; 2019435006@tju.edu.cn (S.L.); 2019435001@tju.edu.cn (Y.L.); liuxin666888999@tju.edu.cn (X.L.); liutao2021@tju.edu.cn (T.L.); fanhj@tju.edu.cn (H.F.); 2General Courses Department, Army Military Transportation University of PLA, Tianjin 300161, China; fu-shao-bo@163.com

**Keywords:** emergency food, supply chain, natural disaster, framework, Delphi approach

## Abstract

A scientific food emergency supply system is helpful for assuring food supplies continuity, improving response efficiency, and reducing disaster losses. However, the framework for a food emergency supply system is currently an understudied area in emergency management post-disaster. In this study, a comprehensive literature review of major databases was performed to identify potential indicators for the emergency food supply system, followed by a two-round modified Delphi with a multidisciplinary expert panel (*n* = 17) to verify the proposed framework. The effective response rate of questionnaires ranged from 94.4% (17/18) to 100% (17/17) and the authority coefficient of experts was 0.88, indicating high positivity and reliability of the experts. Furthermore, the *p*-values of *Kendall’s W* were < 0.01 and the *Cronbach’s α* were > 0.7 for all domains and indicators, indicating a high reliability and validity for the proposed framework. Finally, a consensus was reached on all eight domains and 81 indicators. In conclusion, this study introduced and verified a multidisciplinary framework for the food emergency supply system, which could provide a theoretical basis for emergency responders to make corresponding commands and decisions post-disaster.

## 1. Introduction

Every year from 2011–2020, natural disasters have occurred more than 560 times and were responsible for 25,000 deaths worldwide [1]. Frequent and intense natural disasters pose a great threat to human survival and society, and pose challenges to government capabilities for emergency response and disaster rescue [2]. In the face of disasters, safeguarding human life is the priority for emergency response. Emergency food supply is the dominant response mechanism, which can ensure human survival and basic needs of the affected populations in a disaster situation. In addition, the Humanitarian Charter and Minimum Standards in Disaster Response clearly states that “Access to food and the maintenance of adequate nutritional status are critical determinants of people’s survival in a disaster” [3]. There is a need to concentrate on the construction and development of the emergency food supply system, which can assure emergency food supplies continuity, improve rescue efficiency, and reduce disaster losses.

As a dominant manner for achieving the food supply, the food supply chain is a chain network structure composed of basic links such as material production, collection, reserves, and distribution [4,5,6]. It has aroused growing concerns and research interests due to being highly flexible with the ability to adjust the composition of the supply chain to meet different supply demands as the application situation changes. Especially during disasters, constructing and developing the emergency food supply system from the perspective of the supply chain contributes to maintaining flexibility and resiliency of the emergency food supply system and improving the efficiency of the emergency food supply [7,8,9,10]. However, managing the food supply chain during disaster response is not that straightforward, and changes in disaster environment and society relationship over the last several years have raised the requirement for effective food supply chain management to a new level [11,12,13,14,15]. During most disasters, lack of information, poor coordination, and confused process make it difficult to achieve an effective emergency food supply. Therefore, research on developing the emergency food supply system and improving post-disaster efficiency has become particularly important.

Constructing an emergency food supply system framework based on multidisciplinary might be a good choice, which can lay out the key domains and indicators, construction in a given situation, as well as the relationships between those involved domains and indicators. Furthermore, disaster is characterized by sudden-onset and uncertainty, and the emergency food demand of affected populations changes accordingly. Therefore, it is essential to divide the emergency food supply process into multiple periods (such as the initial period and the middle and late period) according to the emergency management perspective to comprehensively formulate, evaluate, and improve the emergency food supply plan [16,17].

This study proposed a multidisciplinary framework for the emergency food supply system from the perspective of the supply chain, which aimed to realize the coverage of the entire process of emergency food supply at the post-disaster macro level and to find an effective solution to provide emergency food to affected populations who are in need, assist decision-makers in emergency response, and improve the post-disaster emergency food supply efficiency.

## 2. Materials and Methods

### 2.1. Study Design

The modified Delphi study is a systematic approach to transforming individuals’ opinions into a group consensus through a structured questionnaire. Recently, the modified Delphi approach has been widely used across numerous disciplines such as medicine, supply chain management, etc. [18,19,20,21]. In this study, a two-round modified Delphi study was conducted between September and December 2020 to achieve consensus of the multidisciplinary framework for the emergency food supply system, as shown in Figure 1.

### 2.2. Delphi Expert Panel Selection

Studies have shown that the selection and attention of expert panels play crucial roles during Delphi study [22,23]. Therefore, a multidisciplinary expert panel was established based on the following criteria: (1) at least 5 years of working experience as a professional in emergency management, supply chain management, rescue medicine, health service, and other disciplines; (2) at least an intermediate position or a bachelor’s degree in terms of professional title and education, respectively; (3) being experienced in practical emergency supply and in related fields; (4) agreed to participate in the consultation and was willing to provide suggestions or help for this study. Moreover, considering the occurrence of experts’ withdrawal and indicator pool capacity, a total of 18 prospective panelists from different occupations were invited, including government officials, academics, military commanders, and emergency rescuers. All experts were kept confidential to prevent any individual from influencing the panel’s decisions about domains and indicators.

### 2.3. Modified Delphi Procedure

Before conducting the Delphi study, we first identified the core domains and related indicators covered by the emergency food supply system based on the following methods. (i) Literature review: using PubMed, Scopus, Web of Science, CNKI, and Wanfang databases, a systematic literature search was performed between May and July 2020 to identify the content of the initial questionnaire. The search strategies utilized a combination of the following terms: “food”, “emergency food”, “supply”, “supply chain”, “logistic”, “emergency logistic”, “earthquake”, and “natural disaster”. After an initial screening, 2015 articles were identified. Subsequently, the articles were imported to EndNote X9 to remove the duplications. Finally, a total of 133 eligible articles related to the emergency food supply were included and carefully analyzed for the review. With the information assembled through these studies, nearly 140 potential domains and indicators were extracted and used to develop the initial questionnaire, as shown in Table A1. (ii) Policy analysis: potential domains and indicators of the emergency food supply system based on the frequency of relevant words and the relevance with the research contents from the disaster relief emergency plan, regulations, and other relevant documents of national level were extracted [24,25,26,27]. (iii) Focus group discussion: the method was conducted according to regular procedures to supplement and provide content for the initial emergency food supply system [28].

All panelists were sent the questionnaire via mail or e-mail. The questionnaire was composed of three major parts: the first part included section one, which outlined the rationale and purpose of this study; the second part included sections two to four, which were related to demographic information, familiarity with indicators, and scoring criteria of the panelists; and the third part was the importance evaluation of the domains and indicators, which were rated by a Likert-type scale from 1 point (strongly disagree) to 5 points (strongly agree). Moreover, panelists were given the opportunity to suggest additional domains or indicators that they thought would be appropriate for consideration in the next round of the process [29]. The purpose of the first-round consultation included two parts: one part investigated the demographic characteristics of the panelists, such as their gender, education, practical experience, and other relevant information; the self-evaluations on the familiarity of research questions; and the selection basis for each domain and indicator (relying on theoretical analysis, practical experience, reference and intuitive selection); and the other part determined whether domains or indicators in the framework should be added or deleted according to the viewpoints of panelists.

During the second round, some domains and indicators were refined the expression based on the first-round opinion summary, and the first-round results were attached to the questionnaire. All panelists were asked to rerate the importance of the modified domains and indicators and to propose new suggestions for modification until consensus was achieved. Consensus was defined as at least 75% agreement on a domain or an indicator in the current study.

### 2.4. Statistical Methods

Parameters such as the mean importance *(M)*, full score rate *(K_j_)*, and coefficient of variation *(CV)* of each domain and indicator were calculated as the inclusion criteria for domains and indicators. Moreover, the *Cronbach’s α* coefficient and *Kendall’s W* were calculated to verify the reliability and the validity of the proposed framework.

Experts’ positive coefficient of the panel were appraised by calculating the response rate of the questionnaires [30]. The degree of authority of experts reflects the degree of cognition of experts regarding the investigation content, and it is usually expressed by the authoritative coefficient *(Cr)*, which is composed of the basis of expert judgment (*Ca*) and the familiarity with the question *(Cs)* as *Cr* = *(Ca + Cs)*/2 [31]. The *M*, *CV* and *K_j_* were the most used inclusion criteria to judge whether domains and indicators were included. In this study, *M* > 3.5, *CV* < 0.25 and *K_j_* ≥ 50% were regarded as the domain and indicator inclusion criteria for the emergency food supply framework. In principle, only the domains and indicators that met the inclusion criteria were further included and developed, and indicators that failed to meet the criteria were excluded in the following Delphi round.

*Kendall’s W* reflects the degree of experts’ recognition of a given domain and indicator items. A larger *W* indicates a higher degree of coordination and more consistent expert opinions [32]. *Cronbach’s α* coefficient is commonly used to evaluate the internal reliability of the indicator system. An *α* coefficient > 0.70 is considered an acceptable value of framework reliability, as shown in Equation (1) [33].


(1)
$$Cronbach’s\;\;\alpha =\frac{k}{k-1}(1-\overset{n}{\underset{i=1}{\Sigma }}S_{i}^{2}/S_{p}^{2})$$


Weights were calculated based on the mean importance scores for all domains and indicators. Generally, a domain’s weight is the ratio of its mean score to the sum of the mean scores for all domains, and an indicator’s weight is the ratio of its mean score to the sum of the mean scores of all indicators in that dimension [33]. The optimal weights of domains and indicators would help to precisely and accurately construct the emergency food supply system framework. As some stakeholders are involved in the emergency response of only one domain of the system framework (for example, emergency food demand), the indicators and their weights of that domain could also be used individually if separate emergency response of one domain is required.

Data entry and analysis were performed using the IBM Statistical Package for Social Sciences software version 26, and the level of statistical significance was determined as *p* < 0.05.

## 3. Results

### 3.1. Expert Group Statistical Analysis

Eighteen experts were initially contacted to be the Delphi expert panel; of those, 17 experts returned the questionnaires within the specified time. Therefore, after two rounds of consultations, the response rates of the questionnaire were 94.4% (round 1, 17/18) and 100% (round 2, 17/17), respectively. The *Cr* of experts was 0.88 with a *Ca* of 0.94 and *Cs* of 0.81, indicating that the expert panel had higher authority. Table 1 shows the demographic and practical characteristics of the 17 experts.

### 3.2. Revision of the Questionnaire

#### 3.2.1. Round 1

In round 1, all domains were kept the same as the initial domain with the *M* > 3.5, *CV* < 0.25 and *K_j_* > 50%. To make the domains more applicable, the wordings of certain domains were revised to provide greater accuracy. For instance, “emergency security system” was edited to “policy responses to emergency food supply” and “information command system” was edited to “information system of emergency food supply” according to the feedback. Moreover, some experts expressed that the connection between policy responses to emergency food supply (A8) and the emergency food supply system had not been reflected in the questionnaire. Therefore, we only focused on whether the four first-grade indicators (emergency plan (B8.1) (*CV* = 0.06, *K_j_* = 88.2%), emergency management system (B8.2) (*CV* = 0.13, *K_j_* = 70.6%), emergency mechanism (B8.3) (*CV* = 0.08, *K_j_* = 82.4%), and emergency legal system (B8.4) (*CV* = 0.11, *K*_j_ = 82.4%)) among this domain existed, as the specific coverage of each indicator was not in the research scope of this subject based on the research objective. In addition, twelve first-grade and fifteen second-grade indicators were eliminated, seven first-grade and twenty-four second-grade indicators were added, and three first-grade and four second-grade indicators were merged based on the analysis results and expert opinions.

#### 3.2.2. Round 2

In round 2, all domains, first-grade indicators, and the majority of second-grade indicators were kept the same as in round 1 with *M* > 3.5, *CV* < 0.25 and *K_j_* > 50%. Five second-grade indicators “target group casualties (C1.1.2) (*CV* = 0.12, *K_j_* = 47.1%)”, “gender composition of affected populations (C1.2.6) (*CV* = 0.23, *K_j_* = 5.9%)”, “The Ministry of Civil Affairs (C4.2.1) (*CV* = 0.31, *K_j_* = 11.8%)”, “link attribute information (C7.1.3) (*CV* = 0.16, *K*_j_ = 47.1%)”, and “feedback from affected populations (C7.2.7) (*CV* = 0.16, *K_j_* = 35.3%)”, were eliminated based on the inclusion criteria of indicators and the group discussion opinions. The multidisciplinary framework including eight domains, 25 first-grade indicators, and 56 second-grade indicators was achieved after two rounds of expert consultation process. The relationships between domains and indicators are shown in Table 2, and the flow for emergency food supply system is shown in Figure 2.

### 3.3. Weight Allocation of the Framework for Emergency Food Supply System

The weights of domains and indicators were calculated based on the expert consultation results in round 2. As shown in Table 2, the weights of domains and indicators have little difference, which indicated the constructed emergency food supply system framework had good stability.

### 3.4. Degree of Coordination of Experts’ Opinions

The results of the concordance coefficients of expert opinions and significance test are listed in Table 3. The *p*-values of *Kendall’s W* for all domains and indicators were statistically significant (*p* < 0.05).

### 3.5. Reliability of Experts’ Opinions

Seventeen questionnaires were statistically analyzed, and the *Cronbach**’s α* for the eight domains, 25 first-grade indicators and 56 second-grade indicators were all > 0.7, suggesting that the domains and indicators had relatively high internal consistency (Table 4).

### 3.6. Coordination Comparison of the Domains in the Framework

A comparison of eight domains for the final emergency food supply system framework is shown as the Figure 3; and the *M*, *K_j_* and *CV* of eight domains all met the inclusion criteria. However, both emergency food distribution (A5) and emergency food supervision (A6) showed weak coordination compared with other equivalent domains. Therefore, an in-depth study of professionals’ perspectives and views for emergency food supervision and distribution is warranted.

## 4. Discussion

In this study, we designed a multidisciplinary framework for an emergency food supply system based on the supply chain and the emergency management perspective. The framework was divided into eight domains, namely emergency food demand, emergency food reserve, emergency food collection, emergency food supervision, emergency food transportation, emergency food distribution, an information system for emergency food supply, and policy responses to emergency food supply. Furthermore, all domains with almost equal weights indicated that they had similar significance for establishing the multidisciplinary framework. According to the Emergency Response Law of the People’s Republic of China, the State Council, and local people’s governments at or above the county level are the leading administrative organs for emergency response. Therefore, the framework proposed in this paper seeks to complete the emergency food supply in the order of demand prediction, using reserves or other means of collection, emergency supervision, emergency transportation, and emergency distribution on the premise that the emergency food supply is overseen by civil affairs departments at all levels [24].

Emergency food demand is the first step during the emergency food supply. The main function of this domain is to conduct rapid demand forecasting to determine the required types and amount of emergency food needed by the affected populations. Previous studies have mostly emphasized the importance of predicting or evaluating the quantity of emergency food after a disaster, but few studies have focused on the influencing factors of the type of emergency food demanded [2]. Therefore, this domain fully considered the influences of customs and religious beliefs (such as a halal diet, etc.), the season in which disasters occur (such as considering the supply of liquor in winter, etc.), geographical location (such as considering the high quantity of heat of emergency food in plateau-affected areas, etc.), and especially the rescue period on the types of emergency food demand. Influenced by the urgency of emergency food supply in the early period, the type of emergency food in this period was mainly universal food (such as bottled water, instant food, etc.). In the middle and later period, as the disaster situation was alleviated, selective food (such as functional drinks, catering, etc.) and general food (such as vegetables, flour, etc.) were added to the types of emergency food in this period on the basis of universal food. While meeting the basic physiological needs of the affected populations, emergency food supply also meets nutritional needs based on the principle of humanitarian relief [17].

Emergency food reserves have been part of disaster preparedness narratives, and these are maintained to protect access to food for affected populations in the event of a food shortage during emergencies. During the emergency food supply, the geographical distribution and resource configuration of emergency reserves complement each other to ensure the optimal effect on the availability of emergency food [34].

Emergency food collection is a basic domain for the emergency food supply post-disaster, which comprehensively considers the relevant contents of emergency food collection modes, responsible departments, and other factors such as personnel and funds affecting emergency food collection based on the principle of multichannel collection. Although previous studies have identified collection includes using reserves, urgent purchase, direct expropriation, social donations and emergency production, our group reached a consensus that this framework was designed for decision-makers; therefore, ultimately, we ultimately only considered the collection modes led by government (using reserves, urgent purchase, and direct expropriation) in the emergency food supply system. The same is true for the responsible departments and other factors.

Based on the existing literature, a variety of performance metrics have been proposed to build mathematical models for reducing the response time of post-disaster emergency transportation, such as risk, reliability, flexibility, economic loss, and other related measures [35]. Accordingly, in this study, the domain of emergency food transportation focuses on the responsible departments, transportation modes, principles of selecting transportation plans, and other factors such as personnel and transportation tools affecting emergency food transportation based on the emergency responses and optimization principle, and aims to help decision-makers choose the best transportation scheme and shorten the transportation time.

Emergency food distribution is the last step during emergency food supply. Although few previous studies have addressed the distribution problems in the supply process, the participants of the focus group discussion reached a consensus on emergency food distribution for the affected populations. How to distribute emergency food in an orderly way has become a link that cannot be ignored [4]. Therefore, this domain mainly includes the target groups, departments responsible for distribution, and distribution modes in the disaster relief scene.

Emergency food supervision ensures the safety and adequacy (the quality and quantity inspection) of emergency food throughout the entire emergency food supply. Moreover, our expert panel reached a consensus that emergency food should be inspected before transportation and distribution in order to ensure the safety of emergency food supply after a disaster.

In most disasters, information is scarce, which creates disruption in the flow of the food supply chain [36]. Therefore, it is necessary to include an information system when designing an emergency food supply system. The information system can be applied during different periods. By collecting, processing, tracking, and visualizing relevant information during emergency food supply, the system contributes to accessing the current situation, coordinating emergency responses in a timely manner, and finding satisfactory plans for decision-makers.

Policy responses to emergency food supply are indispensable measures to address emergencies and are a particularly important component of national security. Policy responses provide a material basis and technical support for the effective and close implementation of emergency management and rescue work in the field [37]. Therefore, this domain includes the emergency plan, emergency management system, emergency mechanism and emergency legal system of the emergency food supply according to China’s “one plan and three systems” emergency management system to ensure that the post-disaster emergency food supply has laws and evidence to follow.

To the best of our knowledge, the current work has systematically and comprehensively established and demonstrated the framework for the post-disaster emergency food supply system based on existing literature reports. Dividing the emergency food supply system into eight domains from a multidisciplinary perspective will not only clarify the responsibilities of the relevant departments but can also improve the efficiency by standardizing the entire emergency food supply (Appendix A Table A2). In particular, considering that the type of emergency food demanded may vary with the rescue periods, we divided the emergency food supply into two periods, namely the initial period and the middle–late period, to meet the different needs of the affected populations. We believe that the proposed multidisciplinary framework in this paper has comprehensively covered the essential factors for the post-disaster emergency food supply system.

Although the present study design and execution were rigorous, certain limitations still exist. (1) Due to the limitations of the strict inclusion criteria for panelists, the low proportion of females on the expert panel may affect the final results. (2) Considering the generalizability of the results, the framework for the emergency food supply system should be further optimized in order to adapt to more countries and regions. (3) Although the *M*, *K_j_*, and *CV* for the eight domains in the final framework all meet the inclusion criteria, however, the emergency food distribution (A5) and the emergency food supervision (A6) showed a weak coordination compared with other equivalent indicators. Therefore, a detailed framework of the emergency food supply system focused on a certain domain (for instance, A5 and A6) should be demonstrated in the future. (4) The study only focused on the construction of the framework based on the perspective of command decision-makers. It is regrettable that the study did not consider vulnerable populations or people who might need additional support during emergency responses. In future research, these deficiencies will be improved to make research more perfect. (5) According to the formula, *Cronbach’s*
*α* value may higher than the true value when calculating a plenty of indicators.

## 5. Conclusions

In conclusion, this study proposed a multidisciplinary framework for a post-disaster emergency food supply system for emergency decision-makers. The proposal realized the full coverage of participating organizations during emergency food supply at a macro level and could be used as a cognitive tool for emergency workers to make corresponding commands and decisions. The research findings revealed that the framework of the emergency food supply system includes eight domains, which are emergency food demand, emergency food reserve, emergency food collection, emergency food transportation, emergency food distribution, emergency food supervision, an information system, and policy responses during post-disaster emergency responses. Based on a two-round modified Delphi study, a post-disaster emergency food supply system framework containing 25 first-grade indicators and 56 second-grade indicators was ultimately established and the reliability and validity of the framework was well proven using *Kendall’s W* and *Cronbach’s α*.

In this study, the proposed framework has extended the existing literature about the emergency food supply. In particular, the current study can help emergency decision-makers use the system framework to respond more scientifically and effectively to disasters such as strong earthquakes, avoid adverse phenomena such as overlapping responsibilities and redundant responses, and improve the emergency food supply efficiency.

Although the reliability and validity of this framework has been tested, future research is still needed to put the framework into practice to improve the post-disaster emergency food supply efficiency. Furthermore, this study mainly focused on the basic composition of the emergency food supply at the macro level; thus, it is suggested that a more detailed study of the above eight domains can be conducted on the basis of this paper in the future.

## Figures and Tables

**Figure 1 foods-11-01054-f001:**
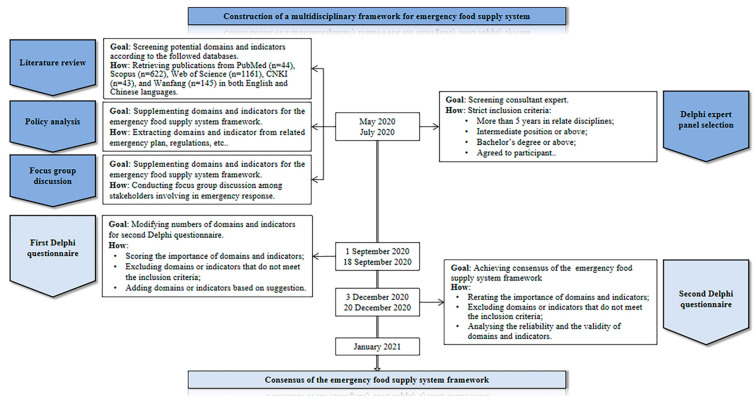
A flowchart depicting the study.

**Figure 2 foods-11-01054-f002:**
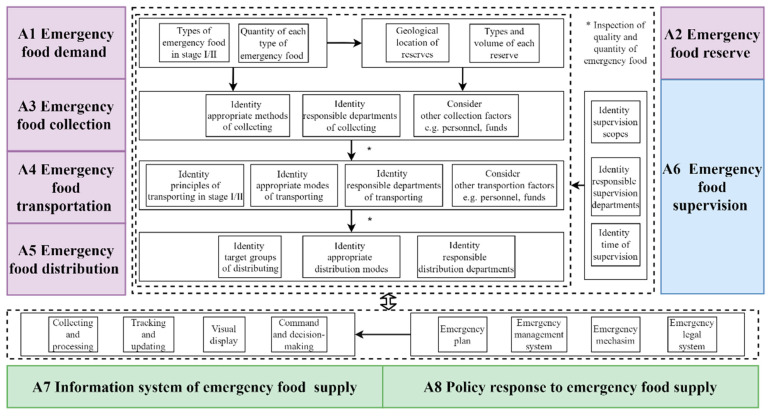
The emergency food supply flow.

**Figure 3 foods-11-01054-f003:**
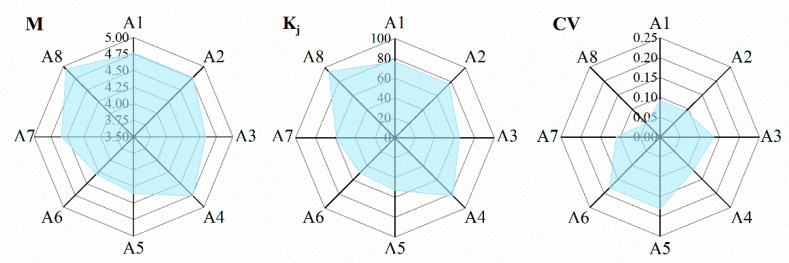
Coordination comparison of emergency food supply domains. Emergency food demand—A1; emergency food reserve—A2; emergency food collection—A3; emergency food transportation—A4; emergency food distribution—A5; emergency food supervision—A6; information system of emergency food supply—A7; policy responses to emergency food supply—A8; mean of importance—*M*; full score rate—*K_j_*; coefficient of variation—*CV*.

**Table 1 foods-11-01054-t001:** Demographics of the panelists (*n* = 17).

Categories	Characteristics	Number	Percentage (%)
Age	31–40 years	3	17.6
	41–50 years	8	47.1
	50~years	6	35.3
Gender	Male	14	82.4
	Female	3	17.6
Academic degree	Bachelor	3	17.6
	Master’s	3	17.6
	Doctor	11	64.7
Professional title	Secondary Senior	7	41.2
	Senior	10	58.8
Working years	5–9 years	9	52.9
	10–19 years	7	41.2
	20–29 years	1	5.9
Research fields	Emergency management	8	47.1
	Logistics management	1	5.9
	Rescue medicine	4	23.5
	Health service	4	23.5
Positions	Management position	2	11.8
	Technical position	15	88.2

**Table 2 foods-11-01054-t002:** The emergency food supply system framework and weightings.

Domains	First-Grade Indicators	Second-Grade Indicators
A1 Emergency food demand (0.130)	B1.1 Demand quantities (0.070)	C1.1.1 Total number of supply target groups (0.036)
		C1.1.2 Disaster relief standards (0.034)
	B1.2 Demand types (0.060)	C1.2.1 Custom and religious belief (0.011)
		C1.2.2 Season of disasters (0.012)
		C1.2.3 Location of disaster area (0.012)
		C1.2.4 Age composition of target group (0.012)
		C1.2.5 Phase of disaster relief (0.012)
A2 Emergency food reserve (0.130)	B2.1 Geographical distribution (0.060)	C2.1.1 Geographical location (0.022)
		C2.1.2 Level of reserve points (0.020)
		C2.1.3 Total number of reserve points (0.021)
	B2.2 Reserve type and volume (0.070)	C2.2.1 Type of reserve (0.032)
		C2.2.2 Reserve quantity (0.034)
A3 Emergency food collection (0.120)	B3.1 Collection methods (0.040)	C3.1.1 Using reserves (0.014)
		C3.1.2 Urgent purchase (0.014)
		C3.1.3 Direct expropriation (0.013)
	B3.2 Responsible departments (0.040)	C3.2.1 National level (0.014)
		C3.2.2 Provincial level (0.014)
		C3.2.3 Municipalities (0.013)
	B3.3 Other factors affecting emergency food collection (0.040)	C3.3.1 Personnel factor (0.022)
		C3.3.2 Funds factor (0.021)
A4 Emergency food transportation (0.130)	B4.1 Responsible departments (0.030)	C4.1.1 Ministry of transport (0.012)
		C4.1.2 NGO * (0.011)
		C4.1.3 Business institutions (0.011)
	B4.2 Transportation modes (0.030)	C4.2.1 Land transportation (0.011)
		C4.2.2 Sea transportation (0.010)
		C4.2.3 Air transportation (0.011)
	B4.3 Principles of selecting transportation plan (0.030)	C4.3.1 Shortest distance (0.010)
		C4.3.2 Least time (0.011)
		C4.3.3 Maximum security (0.011)
	B4.4 Other factors affecting emergency food transportation (0.030)	C4.4.1 Personnel factors (0.015)
		C4.4.2 Transportation tool factors (0.016)
A5 Emergency food distribution (0.120)	B5.1 Target groups (0.040)	C5.1.1 Affected populations (0.021)
		C5.1.2 Rescuer (0.020)
	B5.2 Responsible departments (0.040)	C5.2.1 Relief workers of government (0.019)
		C5.2.2 Volunteers (0.018)
	B5.3 Distribution modes (0.040)	C5.3.1 Direct distribution (0.021)
		C5.3.2 Step-by-step distribution (0.021)
A6 Emergency food supervision (0.120)	B6.1 Supervision scope (0.040)	C6.1.1 Quantity and type (0.021)
		C6.1.2 Quality (safety) (0.021)
	B6.2 Responsible departments (0.040)	C6.2.1 The Food and Drug Administration (0.020)
		C6.2.2 Third party inspection bodies (0.017)
	B6.3 Time of inspection (0.040)	C6.3.1 Before transportation (0.020)
		C6.3.2 Before distribution (0.017)
A7 Information system of emergency food supply (0.120)	B7.1 Supply process information (0.030)	C7.1.1 Basic information of disaster (0.017)
		C7.1.2 Resource point distribution information (0.016)
	B7.2 Tracking and updating information of emergency food supply (0.030)	C7.2.1 Demand information of target groups (0.006)
		C7.2.2 Update information of reserve (0.006)
		C7.2.3 Collection information (0.005)
		C7.2.4 Supervision information (0.005)
		C7.2.5 Transportation information (0.006)
		C7.2.6 Distribution information (0.005)
	B7.3 Information visual display (0.030)	C7.3.1 Visualization of spatial information (0.010)
		C7.3.2 Visualization of model calculation (0.009)
		C7.3.3 Supply dynamic visualization (0.010)
	B7.4 Command and decision-making information (0.030)	C7.4.1 Supply strategy (0.014)
		C7.4.2 Supply options (0.015)
A8 Policy responses to emergency food supply (0.120)	B8.1 Emergency plan (0.030)	
	B8.2 Emergency management system (0.030)	
	B8.3 Emergency mechanism (0.030)	
	B8.4 Emergency legal system (0.030)	

* NGO: Non-Governmental Organizations.

**Table 3 foods-11-01054-t003:** Result of expert opinion coordination degree.

System Frame	Round 1			Round 2		
	W-Value	χ^2^	*p*-Value	W-Value	χ^2^	*p*-Value
Domains	0.131	15.557	0.029	0.156	18.551	0.010
First-grade indicators	0.211	96.812	0.000	0.282	115.122	0.000
Second-grade indicators	0.260	234.602	0.000	0.223	227.256	0.000

**Table 4 foods-11-01054-t004:** Test results of reliability (*Cronbach’s α*).

	Domain	First Grade	Second Grade
Round 1	0.756	0.907	0.957
Round 2	0.705	0.838	0.971

## Data Availability

The data presented in this study are available from the corresponding author [C.C.], and the questionnaire data were not taken from the database, which was publicly available, or relevant sources.

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
