# Peer review of "Establishing a Multidisciplinary Framework for an Emergency Food Supply System Using a Modified Delphi Approach"

_foods, 2022, doi:10.3390/foods11071054_

Round 1

Reviewer 1 Report

The topic of the work is relevant,  timely as it aims at studying the food emergency supply systems from a multidisciplinary perspective.

A reasonable rationale to underpin the research is explained and makes the research interesting. Nevertheless, apart from the statistical methodology, the literature review results could be better focused given the objectives of the research.

The study design is an appropriate approach for this research including the selection of the sample, however, the work does not fully clarify how the structure of the questionnaire is effective in addressing the research questions.  

The eight domains and their elements are described in the discussion chapter however, more detail would be helpful to better highlight the meaning of the weight of the listed indicators as well as their linkage to such domains.  

How the emergency food transportation would be consistent with the existing literature reporting would have been better justified.

Author Response

Response to Reviewer 1 Comments

The topic of the work is relevant, timely as it aims at studying the food emergency supply systems from a multidisciplinary perspective.

Point 1: A reasonable rationale to underpin the research is explained and makes the research interesting. Nevertheless, apart from the statistical methodology, the literature review results could be better focused given the objectives of the research.

Response 1: Thank you for your reminder. The results of the literature review focused on the objectives of the research have been further improved in the Materials and Methods section of the revised manuscript (see line 115-125, page 3). We have provided an additional figure and a additional table to better describe the literature review results (see Figure 1, line 80-97, page 2; Table A1, line 441-442, page 11).

Point 2: The study design is an appropriate approach for this research including the selection of the sample, however, the work does not fully clarify how the structure of the questionnaire is effective in addressing the research questions.

Response 2: We are appreciative of the reviewer’s suggestion. The detailed structure of questionnaire and the expert consultation process have been added and improved in the Materials and Methods section of the revised manuscript (see line 131-149, page 3; line 150-152, page 4).

Point 3: The eight domains and their elements are described in the discussion chapter however, more detail would be helpful to better highlight the meaning of the weight of the listed indicators as well as their linkage to such domains.

Response 3: Thank you for your suggestion. The meaning of the weight of indicators was added in the Materials and Methods section (see line 178-180, page 4), as well as their linkage to such domains have been improved in the Materials and Methods, and the Discussion sections of the revised manuscript (see line 180-183, page 4; line 289-291, page 8).

Point 4: How the emergency food transportation would be consistent with the existing literature reporting would have been better justified.

Response 4: Thank you for your insightful comment. A detailed description of the consistency between the existing literature reporting and emergency food transportation has been added and improved in the Discussion section of the revised manuscript (see line 332-340, page 9).

Reviewer 2 Report

This study “Establishing a multidisciplinary framework for emergency food supply system using a modified Delphi approach” seeks to propose a multidisciplinary framework for the emergency food supply system from a supply chain perspective, which aimed to achieve the coverage of the entire emergency food supply process at the post-disaster macro level and find an effective solution for emergency feeding to the affected populations.

The topic is both relevant and important. Nevertheless, we propose below some elements that can help to improve it:  

1, Reference 18. did not address the modified delphi as mentioned in "2.1. Study Design".

2. In section "2.4. Statistical Methods" the authors present "Mj ≥ 50%" here is it "Mj" or "Kj"?

3. Did the authors really have a choice of panel? Actually choosing 18 to keep 17 seemed more like a designation than a choice! It might be better to start with a larger number of experts, set up inclusion criteria to finally arrive at a rather coherent group. See the inconsistencies in the next remark (N°4)

4. Also, there are some imbalances in the presentation table of the panelists: Demographics of the panelists at the following levels:

- Gender (14 3)

- Academic degree (3/3 11)

- Working years (9/7 1)

- Positions (15 2)

Doesn't this imbalance have an impact on the quality of the responses and subsequently on the results of the study?

5. The reliability test in Table 4 (Cronbach's α) may call into question the entire approach. In fact, it is 0.957 for the first round and 0.971 for the second round. Therefore, a Cronbach's α greater than 0.9 is only desired in exceptional cases, as it may indicate that the statements are too similar, thus reducing the real reliability of the scale.

6. Figure 2 cannot have such a long title. It is strongly recommended to summarize it.

7. Have the results obtained been applied to a real field situation. If not, are the data used from the data collected in a real situation?

8. If the data is from the Republic of China, one may wonder about the generalizability of the results to other countries.

9. The authors claim that a comprehensive literature review of major databases was conducted to identify potential indicators of the emergency food supply system. However, only the names of the databases and the filters used were presented, but they did not present the results for this section.

10. There is redundancy between the data in some tables in the main text and some items in the appendices.

11. Summary tables could be very useful to enhance the work done in the presentation of the data collected in the databases (at the beginning) and in the discussion section (at the end).

Author Response

Response to Reviewer 2 Comments

This study “Establishing a multidisciplinary framework for emergency food supply system using a modified Delphi approach” seeks to propose a multidisciplinary framework for the emergency food supply system from a supply chain perspective, which aimed to achieve the coverage of the entire emergency food supply process at the post-disaster macro level and find an effective solution for emergency feeding to the affected populations.

The topic is both relevant and important. Nevertheless, we propose below some elements that can help to improve it:

Point 1: Reference 18. did not address the modified Delphi as mentioned in "2.1. Study Design".

Response 1: Thanks for your suggestion, we have updated Reference 18 in the References section of the revised manuscript (see line 478-479, page 15).

Point 2: In section "2.4. Statistical Methods" the authors present "Mj ≥ 50%" here is it "Mj" or "Kj"?

Response 2: Many thanks. It is “Kj ≥ 50%” in the section “2.4. Statistical Methods”, “3.2.1. Round 1” and “3.2.2. Round 2”, and has been modified in the corresponding sections of the revised manuscript.

Point 3: Did the authors really have a choice of panel? Actually choosing 18 to keep 17 seemed more like a designation than a choice! It might be better to start with a larger number of experts, set up inclusion criteria to finally arrive at a rather coherent group. See the inconsistencies in the next remark (N°4)

Response 3: Your suggestion is fine. In our study, the inclusion criterias for panelists were strict to ensure the typical and authority of the experts, including enough working and practical experience in related disciplines or fields, a higher professional title or education background, etc. (see line 101-106, page 3). Because one of the 18 eligible experts did not return the questionnaire within the specified time in Round 2, only 17 experts were kept finally in our study, which is a normal range of variation in the number of experts or consultants (see line 190, page 4).

Point 4: Also, there are some imbalances in the presentation table of the panelists: Demographics of the panelists at the following levels:

- Gender (14 3)

- Academic degree (3/3 11)

- Working years (9/7 1)

- Positions (15 2)

Doesn't this imbalance have an impact on the quality of the responses and subsequently on the results of the study?

Response 4: We are appreciative of the reviewer’s suggestion. It is undeniable that the gender bias of the expert group is an obvious limitation in this study, and that has been addressed in the Discussion section of the revised manuscript (see line 381-382, page 10). Previous studies have demonstrated that panelists with higher degree, more professional skills, and moderate working years may have a positive impact for the consultation results (e.g. a rapid and efficient recovery of valid questionnaires) based on the research objective. (ref. Devaney L, Henchion M. Who is a Delphi 'expert'? Reflections on a bioeconomy expert selection procedure from Ireland[J]. Futures, 2018, 99:45-55.)

Point 5: The reliability test in Table 4 (Cronbach's α) may call into question the entire approach. In fact, it is 0.957 for the first round and 0.971 for the second round. Therefore, a Cronbach's α greater than 0.9 is only desired in exceptional cases, as it may indicate that the statements are too similar, thus reducing the real reliability of the scale.

Response 5: Thank you for your suggestion. Cronbach’s α was utilized as a measure of internal consistency of each part, and α values of 0.7 to 0.8 were regarded as satisfactory. (ref. Bland JM, Altman DG. Statistics notes: Cronbach’s alpha. BMJ. 1997, 314: 572–573.) In this study, Cronbach’s α in all domains and indicators were all > 0.7, which can been regarded as satisfactory for the internal consistency of domains or indicators. However, from the perspective of mathematical theory, Cronbach’s α value may be higher than the true value when calculating a large number of indicators according to the formula. The potential limitation was added in the Discussion section of the revised manuscript (see line 394-395, page 10).

Point 6: Figure 2 cannot have such a long title. It is strongly recommended to summarize it.

Response 6: Many thanks. The title of Figure 2 has been summarized in the revised manuscript (see line 278-282, page 8).

Point 7: Have the results obtained been applied to a real field situation. If not, are the data used from the data collected in a real situation?

Response 7: Thank you for your suggestion. The results of our study have not been applied to a real field situation. Nevertheless, the domains and indicators of the framework were demonstrated basing on a comprehensive literature review, the study of a number of real cases, deliberative policy analysis, and the opinions of multidisciplinary panelists, which is conducive to achieving high applicability of the constructed framework in real field situation.

Point 8: If the data is from the Republic of China, one may wonder about the generalizability of the results to other countries.

Response 8: You have raised an important point. Although this study is based on the China’s basic national conditions and the emergency response laws, the proposed system framework is applicable to other developing countries, as long as appropriate modifications are made in combination with the corresponding national background. Nevertheless, we have added the contribution limitations of the results in the Discussion section of the revised manuscript (see line 382-384, page 10).

Point 9: The authors claim that a comprehensive literature review of major databases was conducted to identify potential indicators of the emergency food supply system. However, only the names of the databases and the filters used were presented, but they did not present the results for this section.

Response 9: Thank you for your reminder. The results of the literature review focused on the objectives of the research have been further improved in the Materials and Methods section of the revised manuscript (see line 115-125, page 3). We have provided an additional figure and a additional table to better describe the literature review results (see Figure 1, line 80-97, page 2; Table A1, line 441-442, page 11).

Point 10: There is redundancy between the data in some tables in the main text and some items in the appendices.

Response 10: Thank you for your reminder. It has been modified for the redundant tables and some items information between main text and appendices in the revised manuscript (see Table A2, line 442, page 12).

Point 11: Summary tables could be very useful to enhance the work done in the presentation of the data collected in the databases (at the beginning) and in the discussion section (at the end).

Response 11: Thank you for your suggestion. A flowchart depicting the study has been added in the Materials and Methods section of the revised manuscript (see Figure 1, line 80-97, page 2).

Round 2

Reviewer 2 Report

The authors have made a commendable effort and have responded to the majority of the comments made in the first reading.
However, two small remarks/suggestions remain:
- Remark N°6: Figure N°2 (current N°3). It is always possible to envisage a shorter title and at the same time significant.

Remark 7: Take into consideration the need to apply the fieldwork (not necessarily in this work, but in future work).  In fact, no other approach (documentary analysis, case studies...) can replace this fieldwork.